# Choline and Fish Oil Can Improve Memory of Mice through Increasing Brain DHA Level

**DOI:** 10.3390/foods12091799

**Published:** 2023-04-26

**Authors:** Jin Li, Yaqiong Jian, Ruonan Liu, Xianfeng Zhao, Jiangyi Mao, Wei Wei, Chenyu Jiang, Lina Zhang, Yi Wang, Peng Zhou

**Affiliations:** 1State Key Laboratory of Food Science and Technology, Jiangnan University, Wuxi 214126, China; 2School of Food Science and Technology, Jiangnan University, Wuxi 214126, China; 3Danone Open Science Research Center for Life-Transforming Nutrition, Shanghai 200135, China

**Keywords:** n-3 fatty acids, docosahexaenoic acid, choline, blood–brain barrier, learning and memory

## Abstract

Docosahexaenoic acid (DHA) is highly enriched in the brain, and is essential for normal brain development and function. However, evidence suggests that currently used supplements, such as fish oil, do not significantly increase brain DHA levels. Therefore, this study aimed to investigate whether combined fish oil and choline supplementation could affect the type and enrich the content of DHA in the brain. The results revealed that the combined intake of fish oil and choline upregulated the expression of key transporters and receptors, including MFSD2A, FATP1, and FABP5, which increased the uptake of DHA in the brain. Additionally, this supplementation improved the synthesis and release of acetylcholine in the brain, which, in turn, enhanced the learning and memory abilities of mice. These findings suggest that the combined intake of fish oil and choline improves the bioavailability of DHA in the brain.

## 1. Introduction

Omega-3 polyunsaturated fatty acids (n-3 PUFAs), such as docosahexaenoic acid (DHA), are important structural components of nerve cell membranes and have important functions in various membrane binding processes [1]. The lack of DHA in the brain of developing newborns or children will likely result in neurological aspects such as cognitive impairment and poor memory [2]. For adults or the elderly, brain DHA deficiency is associated with a variety of neurological disorders, including Alzheimer’s disease, Parkinson’s disease, schizophrenia, and depression [3]. Since the synthesis of DHA is very low in the brain, the brain DHA is mainly supplied from blood plasma [4]. The blood plasma DHA in plasma is directly affected by some dietary supplements. Fish oil, which is rich in n-3 PUFA, mostly in the form of triglycerides (TAG), is the most common dietary supplement for DHA intake [5]. Dietary supplementation with n-3 PUFA can rapidly increase the level of n-3 PUFA in the blood [6]. DHA in plasma exists in different forms, such as nonesterified DHA (NE-DHA) and some esterified DHA such as TAG-DHA, phosphatidylcholine-DHA (PC-DHA), lysophosphatidylcholine-DHA (LPC-DHA), etc. [7]. The transportation of DHA from plasma to brain is regulated by the blood–brain barrier (BBB), which is the barrier between plasma and brain cells formed by the walls of brain capillaries and glial cells, to enter the brain [8]. However, current research shows that DHA mainly crosses the blood–brain barrier in the form of NE-DHA or LPC-DHA and enters the brain [9]. The NE-DHA crossing through BBB depends on many proteins, such as fatty acid binding proteins (FABPs), fatty acid transport proteins (FATPs), and long-chain acyl coenzyme a synthases (ACSLs), among others [10]. However, the crossing of LPC-DHA through the BBB requires a specific transporter protein, the major facilitator superfamily (MFSD2A) [11,12]. Several studies have shown that single supplementation with fish oil (DHA dietary supplements) does not significantly enrich DHA in the brain [13,14,15] because DHA is released in the form of free DHA or monoacylglycerol during TAG digestion and exists in plasma as TAG, which is not well absorbed by the brain [15]. The content of DHA in the brain is highly correlated with the normal development and functional maintenance of the central nervous system, and DHA levels in the brain are linked to memory [2]. The effect of ingesting fish oil alone on improving memory in mice was also not significant [16].

Choline, similar to DHA, is an essential nutrient and must be obtained from food [17]. Choline and DHA are constitutive body components, linked via PC metabolism. PC-DHA is enriched in brain grey matter, is present in all tissues, and is tightly regulated in tissues [18]. Choline is a component of all biological membranes and is a precursor to acetylcholine in cholinergic neurons [19,20]. Choline intake in adults can have an impact on cognitive performance because choline acts as a precursor to PC, a major component of neuronal cell membranes [21]. Choline can biosynthesize PC in two ways: (1) choline biosynthesizes PC via the CDP–choline cycle or the Kennedy cycle, and (2) choline synthesizes PC via PEMT [21,22]. It has been shown that PC-DHA concentrations in plasma and erythrocytes significantly increased in subjects receiving a diet with choline and DHA, and decreased in subjects on a choline-deficient diet [23,24]. However, still unknown are the changes in the different molecular forms of DHA in brain tissue after intake of fish oil (unsaturated fatty acid nutritional supplement) and choline and whether the combined intervention of fish oil and choline can increase DHA levels in the brain. Choline is also a precursor needed to form acetylcholine, which is a neurotransmitter that controls memory and muscle movements, and its synthesis requires the participation of acetylcholine transferase [25]. The release of neurotransmitters is accompanied by the fusion of vesicles with the cell membrane. Syntaxin3 and SNAP-25 are essential proteins in the process of vesicle transport and are responsible for the final fusion of vesicles with the cell membrane. After the vesicle fuses with the cell membrane, neurotransmitters are released from the vesicle into the synaptic cleft, where these neurotransmitters bind to receptors in the posterior membrane and transmit information to the next neuron; we call this synaptic transmission. It is also an important process of synaptic plasticity and is closely related to higher cognitive functions such as learning and memory [26].

There is little known about the impact of fish oil on healthy middle-aged male mice. To date, male studies have primarily been completed in disease models and young mice. It is now widely accepted in the industry that aging can lead to memory loss and that early intervention can be made through diet, so more attention is now being paid to the effects of aging on middle-aged people in their forties due to aging. In the current study, experiments were conducted primarily on middle-aged and older mice to study the effects of DHA intake on memory in this age group. Eight-month-old mice eventually reach 10 months of age or older after feeding and fall into the category of middle-aged mice equivalent to the 40-year-old stage in humans. Therefore, this study used 8-month-old mice to investigate whether the combined intake of fish oil and choline significantly improved memory in mice and to investigate the mechanism of why the combined intake of fish oil and choline affected memory in mice.

## 2. Materials and Methods

### 2.1. Materials

On the basis of AIN93M standard feed, palm stearin was used instead of soybean oil, and the rest basically met the requirements of the national standard GB14924.3-2010 of the People’s Republic of China. It was purchased from Jiangsu Synergy Pharmaceutical Bioengineering Co., Ltd. (Nanjing, Jiangsu, China) (Table 1). MFSD2A, FATP1, FABP5, Syntaxin3, SNAP-25 and β-actin primers were synthesized by Genewiz Biotechnology Co., Ltd. (Suzhou, Jiangsu Province, China).

### 2.2. Composition of F, C, and FC

In this experiment, two types of fish oil powder raw materials were used, and the two fish oil powders were mixed in a certain proportion to form fish oil with a dosage ratio of DHA and EPA of 4:1. Our fish oil is from Novosana; we performed a peroxide test prior to the experiment and the result was 8.3 meq/kg, which met the experimental criteria, and then we added 0.5% carboxymethyl starch solution. The C group (choline group) was prepared using choline bitartrate plus 0.5% carboxymethyl starch solution. FC (complex of fish oil and choline) was prepared from fish oil powder and choline bitartrate plus 0.5% carboxymethyl starch solution, and the dose of DHA and choline in fish oil was 3:1 by mass. The control group was provided with only 0.5% carboxymethyl starch solution. The composition of each group is shown in Table 2. The doses of fish oil and choline used in this experiment were based on the recommended intake levels for humans, which are commonly available in the market. After being converted into appropriate doses for mice, they were administered via intragastric gavage.

### 2.3. Animals and Experimental Design

Animal experiments were approved by the Animal Care and Use Committee of the Institute of Animal Nutrition, Jiangnan University, and carried out in accordance with the National Laboratory Animal Welfare Ethics Review Guidelines. To study the effect on midlife stage production, we used 8-month-old mice. All animals were raised according to the standard, and after a 7-day acclimatization period, the mice were randomly divided into 4 groups according to body weight (BW), with 10 mice in each group. Sample size for experiments was decided based on power calculation from previous experiments and experiences. They were fed custom feed throughout the rearing period. The customized feed is shown in Table 1. Oral doses for mice are shown in Table 3. The body weight of each mouse was measured weekly. Behavioral experiments were performed after daily gavage for 30 days. The animal experiment process is shown in Figure 1b.

### 2.4. Sample Collections

Two months after the intervention, the mice were euthanized. Blood was taken by removing the eyeball, centrifuged at 3000 rpm for 15 min, and the supernatant was drawn. The plasma obtained was stored in a refrigerator at −80 °C. Whole brain hemispheres were removed and fixed in paraformaldehyde solution. The brain tissue and visceral tissue of the mice were collected, and the cerebral cortex, hippocampus, liver, and kidney were weighed and placed at −80 °C for long-term storage.

### 2.5. Water Maze Test

The water temperature was about 20 °C in the labyrinth lane at a depth of 9 cm. The single training time was limited to 2 min; if mice did not reach the end point within 2 min, their reaching time was recorded as 2 min. The mice were placed near the ladder before the first training and made to climb up automatically 3 times. Before each subsequent training, the mice were placed near the ladder with their backs facing the stairs and made to climb up once automatically. The experiment was carried out in stages, and the distance was gradually lengthened depending on the learning performance of the animals. The endpoint test was performed by placing the mice at the starting point and recording the time required to reach the endpoint, as well as recording the number of errors that occurred in between. For each training session, mice that did not reach the end point within 2 min were guided to the end point and up the stairs from the end point for the purpose of training. During each training or test, the head was turned towards the starting point. Finally, the total number of errors and the total time to reach the end point were calculated for each group of animals during the 5 training and testing sessions (entering any of the blind ends once counted as one error) [27].

### 2.6. Step-through Passive Avoidance Test

Training started the day after the last sample administration (or 1 h after a single administration). During the experiment, the mice were placed into the open test box with their faces facing away from the hole, and the timer was started at the same time. Mice moved from a light room to a dark room, where they received electric shocks. This period was recorded as the latency period. The number of shocks was recorded as the number of mouse errors. We took out the mice, calculated the time required for each mouse to enter the dark room from the time it was placed in the light room, and trained for 5 min. The experiment was repeated after 24 h or 48 h, and the incubation period of each animal entering the dark room and the number of electric shocks within 5 min were recorded. Memory decline experiments were performed 5 days after training was stopped [27].

### 2.7. Targeted Lipidomic Analysis

According to previous research with modifications [28], all samples were run on a Sciex LC coupled to an AB Sciex 5500 Triple Quadrupole Ion Trap Mass Spectrometer (Q-TRAP-MS) in positive and negative electrospray ionization (ESI) modes. Instrument control and data integration were performed using Analyst Software Version 1.6.2. We used different standards for each lipid and used the mixed-label product SPLASH Lipidomix (330707) from Avanti Polar Lipids.

### 2.8. qRT-PCR Analysis

The mRNA expression of genes involved in the release of fatty acid transport neurotransmitters, such as MFSD2A, FATP1, FABP5, Syntaxin3, and SNAP-25, was measured using qRT-PCR. β-actin mRNA was used as an internal control. Total RNA was extracted from cerebral cortex and hippocampus (30 mg) using FastPure Cell/Tissue Total RNA Isolation Kit V2 columns according to the instructions, followed by reverse transcription into cDNA. The total reaction volume was 10 μL, and the cycle amplification was carried out with a fluorescent PCR instrument. All data were normalized to the reference mRNA of β-actin, and relative quantification was performed using the 2^−ΔΔCt^ method. All primers used are shown in Table 4.

### 2.9. Western Blot Analysis

Cerebral cortex and hippocampus weighing 30 mg were homogenized with pickaxe beads in a homogenizer, and proteins were extracted with RIPA. Protein concentration was checked after centrifugation. After denaturation at high temperature, they were separated by sodium dodecyl sulfate-polyacrylamide gel electrophoresis on a 12% separating gel. They were then transferred to cellulose membranes, blocked with blocking solution, and incubated overnight at 4 °C with primary antibodies against MFSD2A, FATP1, FABP5, Syntaxin3, SNAP-25, or β-actin. The next day, we added oxidase-conjugated goat anti-rabbit antibody and IgG (H + L) secondary antibody, and incubated for 2 h in the dark. Bands were finally visualized, and all data normalization was performed based on an internal control protein of β-actin.

### 2.10. Determination of ChAT Activity and ACh Content in the Brain

The weight of the tissue was accurately weighed, the tissue homogenate was prepared with normal saline in a weight-to-volume ratio, and the supernatant was collected by centrifugation for determination. The contents of choline acetyltransferase (ChAT) and acetylcholine (ACh) in the cerebral cortex and hippocampus were determined in strict accordance with the kit instructions.

### 2.11. Immunocytofluorescence and Immunohistochemistry

The fresh mouse brain tissues (left half brain and right half brain) were taken and fixed in paraformaldehyde, dehydrated and then embedded in paraffin wax, and then the paraffin wax was cut into 4 μm sections. Paraffin sections were dewaxed and rehydrated, and then incubated with Claudin-5 primary antibody and then secondary antibody, respectively. CY3-TSA was added and then treated with microwave heating. Similarly, MFSD2A primary antibody was used to complete the above operation and then microscopic examination was performed. MFSD2A-positive markers were red, Claudin-5-positive markers were green, and nuclei were blue. Immunohistochemistry experimental steps were performed with primary antibody of FATP1 and FABP5 followed by incubation with secondary antibody. Finally, the tissues were stained with DAB for color development, restained with hematoxylin, and then the images were collected and analyzed under a microscope. The nucleus of hematoxylin-stained tissue was blue, and the positive expression of DAB was brownish yellow.

### 2.12. Statistical Analysis

All data were assessed for differences between groups using one-way ANOVA, followed by statistical analysis of differences using SPSS software for social sciences version 22.0 (SPSS Inc., Chicago, IL, USA). The statistical power (power) of the behavioral experiments was calculated by G*Power 3.1.9.2 software with an α-error probability value of 0.05. Bars with a common letter superscript are not significantly different from each other; *p* < 0.05 indicates a significant difference, and *p* < 0.01 indicates an extremely significant difference.

## 3. Results

### 3.1. Effects of F, C, and FC on Mouse Body Weight, Liver, and Kidney Weight

As shown in Figure 1a, there was no significant difference in body weight among fish oil, choline, and FC groups. Similar results were also found in the weight of liver and kidney, as shown in Figure 1b,c. Therefore, the intake of fish oil or choline did not affect the changes in body weight and liver and kidney of mice.

### 3.2. Effect of F, C, and FC on Improving Memory in Mice

Behavioral tests are commonly used in nonclinical effectiveness studies to evaluate the impact of drugs or other interventions on learning and memory function in animals. The behavioral experiments we used included the step-through passive avoidance experiment and water maze experiment, which are currently industry-recognized and commonly used tests for detecting learning–memory abilities. Sample size for experiments was decided based on power calculation from previous experiments and experiences. Statistics and calculations of behavioral experimental results were based on previous studies with modifications [27,29,30]. In behavioral experiments, the longer the latency of the mouse and the lower the number of errors of the mouse, the better its learning and memory ability. In this study, we measured the latency and number of errors in these tests (Table 5 and Table 6). As demonstrated in Table 5, during the training test, there was no significant difference in the total time taken by the mice in each group to reach the end point, but the mice that ingested fish oil and choline took less time compared to the control group. Mice in the FC group made significantly fewer errors in the water maze training than the control group (*p* < 0.05, power = 0.976). Similarly, in the water maze test experiment, the number of errors in the FC group of mice was significantly lower than the control group (*p* < 0.05, power = 0.8336). In the second training test (Table 6), there was a significant difference in the number of errors between the FC group and the F group. Interestingly, although there was no significant difference in the time spent in the water maze memory test among all groups, the mice in the FC and choline groups had a lower number of errors compared to the other groups. In the memory test of the step-through passive avoidance experiment, the mice in the FC group had fewer errors and the longest latency compared to the other groups. In the step-through passive avoidance test experiment, the number of errors in FC mice was significantly lower than in the control group (*p* < 0.05, power = 0.951) (Table 6).

### 3.3. Effects of F, C, and FC on DHA Content in Mouse Brain

Lipid extracts from mouse brain were analyzed by LC-MS/MS for the nine main lipid types in mouse brain tissue, and the total DHA content in mouse cerebral cortex and hippocampus (Figure 2a,b) were examined separately, as well as the content of different types of DHA in mouse brain, plasma and liver; these included NE-DHA, DAG-DHA, TAG-DHA, PC-DHA, SM-DHA, PE-DHA, PG-DHA, and LPC-DHA (Figure 2c–e). We observed a significant increase in the total DHA content in the cerebral cortex and hippocampus of mice in the FC group relative to the control group. The DHA content in the cerebral cortex and hippocampus of mice in F group was slightly higher than that in the control group, but there was no significant difference. In brain, plasma, or liver tissues, the FC group of mice had higher levels of each type of DHA than the other groups. In brain and liver tissues, the content of PC-DHA was significantly higher in the FC group than in the other groups. In plasma and liver tissues, the levels of TAG-DHA and LPC-DHA were significantly higher in the FC group than in the other groups.

### 3.4. Effects of F, C, and FC on Fatty Acid Transporters in Mouse Brain

Our data showed that the mRNA levels as well as protein levels of MFSD2A in the cortex of the FC group were significantly higher than those of the control group, and there were no significant differences between F and C groups relative to the control group, while the mRNA levels and protein levels of MFSD2A in the cortex of the F group were relatively higher than those of the control group (Figure 3a,d). Interestingly, the mRNA levels and protein levels of cortical FATP1 in the FC group did not change significantly compared to the control group (Figure 3b,e). We found that the mRNA level of cortical FABP5 was significantly upregulated in the FC group compared to the control group, but there was no difference in the protein level (Figure 3c,f).

In addition to the cortex, we found similar trends in hippocampal tissues of mice, that the mRNA levels of MFSD2A and FABP5 as well as the protein levels in the hippocampus of the FC group were significantly higher than those in the control group (Figure 3g,i,j,l). Interestingly, the mRNA and protein levels of FATP1 in the hippocampus of each group were not significantly different (Figure 3h,k).

### 3.5. Immunofluorescence Double Staining to Detect the Positive Expression and Localization of MFSD2A in the Brain

Double-label immunofluorescence staining showed that MFSD2A immunoreactive cells produced red fluorescence, and the positive product was mainly located in the cell membrane, while the nucleus was not immunoreactive. Claudin-5 immunoreactive cells produced green fluorescence and the positive product was also mainly located in the cell membrane, and immunofluorescence localization indicated that MFSD2A and claudin-5 were specifically expressed in the brain microvessels that constitute the BBB endothelial cells, which can be clearly observed in sagittal sections of mouse brain (Figure 4a). We also found positive expression of MFSD2A and Claudin-5 in the choroid plexus vessels of the fourth ventricle (Figure 4b). To more precisely analyze the difference in MFSD2A expression in the mouse cortex, hippocampus, and vascular choroid plexus, we performed a semiquantitative analysis of fluorescence intensity for this. We found that the mean fluorescence intensity of MFSD2A in the cortex and hippocampus of the FC group was significantly different compared to the control group. Interestingly, the mean fluorescence intensity in the cortex and hippocampus of mice ingesting fish oil tended to increase, but there was no significant difference (Figure 5a,b). MFSD2A was positively expressed in the choroid plexus of the fourth ventricle in all groups of mice, but there was no significant difference between the groups (Figure 5c).

### 3.6. Effects of F, C, and FC on Cholinergic Neuron System in Mouse Brain

Our data showed that the mRNA levels of SNAP-25 and Syntaxin3 in the cerebral cortex of mice in the fish oil and FC groups were significantly higher than those of mice in the control group (*p* < 0.01) (Figure 6a,b). Interestingly, the mRNA level of SNAP-25 was highest in the choline group (Figure 6a), but the protein levels of SNAP-25 and Syntaxin3 were not statistically different in the cerebral cortex of mice in each group (Figure 6c,d).

We compared the hippocampus with the cortex and found that the mRNA levels of SNAP-25 and Syntaxin3 in the hippocampus of mice in the FC group were significantly higher than those of mice in the control group (Figure 6e,f), and the protein levels of SNAP-25 in the hippocampus of mice in the FC group were significantly higher than those in the control group (*p* < 0.05), while the protein levels of Syntaxin3 did not differ between groups (Figure 6g,h).

We further compared the content of ACh and the level of ChAT activity in the cerebral cortex of mice and the content of ACh and the level of ChAT activity in the hippocampus between different groups (Figure 7a–d). The experimental results showed that the content of ACh in the cerebral cortex and hippocampus of mice in the FC group was significantly higher than that in the control group (*p* < 0.05), by nearly 1.5 times (Figure 7a,c). The ChAT activity in the cerebral cortex of the C group and the FC group was significantly different from the control group (*p* < 0.01) (Figure 7b), but we found no difference in the hippocampus (Figure 7d).

### 3.7. Immunohistochemical Detection of the Positive Expression of FATP1 and FABP5 in the Brain

We found that FATP1 was distributed in the cerebral cortex, hippocampus, and choroid plexus of the fourth ventricle in all groups of mice, and the immunoreactivity of FATP1 was clearly observed in the cytoplasm and cytoplasmic membrane (Figure 8).

Interestingly, in all groups of mice, positive immunostaining for FABP5 was observed in the cerebral cortex and hippocampus, but not in the choroid plexus of mice. From the figure, it can be observed that the immunoreactivity of FABP5 was seen in the cytoplasm of endothelial cells, with the hippocampus of the FC group of mice showing a higher intensity of staining (Figure 9).

## 4. Discussion

In the current study, we found that combined fish oil and choline supplementation can increase brain uptake of DHA from fish oil by upregulating the expression levels of the transporter proteins MFSD2A, FATP1, and FABP5, facilitating the passage of DHA across the blood–brain barrier to the brain. The combined supplementation of fish oil and choline was further verified by behavioral experiments to have significant effects on memory in mice.

The content of DHA in the brain was highly correlated with the normal development and functional maintenance of the central nervous system [2]. However, previous research showed that dietary supplements such as fish oil intake of DHA do not enrich DHA in the brain, because DHA is mainly digested and absorbed in the form of TAG, while the molecular form of DHA ingested by the brain is LPC or NE-DHA [31]. However, it is still debated as to which of these two molecular forms of DHA is the primary donor of DHA to the brain. Chuck T. Chen directly compared the uptake of the two in the brain and found that the brain entry rate of NE-DHA was 10 times higher than that of LPC-DHA [7], and Dhavamani Sugasini showed that LPC-DHA was the main source of DHA enriched in the brain [16].

LPC-DHA in plasma can be converted from PC-DHA catalyzed by phosphatidylcholine-cholesterol acyltransferase. Choline is, in turn, a synthetic precursor of PC, which is synthesized in the body via the Kennedy pathway or the PEMT pathway. Circulating LPC-DHA levels are regulated by the supply of dietary DHA and choline [21]. Previous studies found that plasma levels of PC-DHA were greater in the group that consumed choline combined with DHA than the group that consumed DHA alone [32]. Therefore, the combined intake of fish oil and choline would increase circulating levels of PC-DHA and levels of LPC-DHA, thus affecting the absorption of DHA in the brain. This is in line with our results showing that the combined intake of fish oil and choline significantly increased PC-DHA and LPC-DHA levels in the brain (Figure 2c), whereas fish oil intake alone was not effective in increasing LPC-DHA levels in the mouse brain (Figure 2c). The liver is the main site of phospholipid synthesis such as PC, and we found that more PC-DHA and other lipids were synthesized in the livers of the FC group mice, and these different types of DHA were further released into the plasma for uptake by the brain. Furthermore, we found that the combined intake of fish oil and choline significantly increased the amount of free DHA in the brain, while the total DHA content in the brain was also significantly higher than in the control group (Figure 2a,b). The absorption of DHA by the brain requires the participation of fatty acid transporters [9,10]. The fatty acid transporter MFSD2A on the blood–brain barrier can transport DHA in the form of LPC across the blood–brain barrier into the brain, while FATP1 and FABP5 transport free DHA into the brain. Fatty acids are natural ligands for the oxidase proliferator-activated receptor (PPAR), and previous studies found that DHA itself can affect the activation of PPAR, which may be involved in the regulation of FATP1 and FABP5, thereby upregulating the expression of FATP1 and FABP5 [33,34]. Moreover, the protein expression of MFSD2A was likewise affected by the mRNA expression of PPAR-γ (one of the types of PPAR) [35]. We believed that ingestion of fish oil and choline might activate PPAR, thereby upregulating the expression level of fatty acid transporter proteins (MFSD2A, FATP1, and FABP5, Figure 3, Figure 4 and Figure 8), which may promote DHA uptake by the brain and thereby increase the level of DHA in the brain. Further studies are still needed to confirm this hypothesis.

Higher level of DHA in the brain plays a crucial role in brain cognitive function and learning memory capacity. The improvement of fish oil combined with choline in the mice model may be related to either higher levels of DHA in the brain or the consumption of choline. Previous studies showed that neuronal synaptic plasticity was fundamental to learning and memory. Choline and DHA could act as precursors for the synthesis of neuronal synaptic membrane phospholipids (e.g., PC), which stimulate membrane formation and function [36,37]. The intake of fish oil and choline increased the level of DHA in the brain and promoted the synthesis of phospholipids in the brain, which affected neuronal function and thus improved memory in mice. Choline was also a precursor for the synthesis of the neurotransmitter Ach, which played a critical role in cognition, supporting perception, attention, and learning and memory [25,38]. The basis of the neurotransmitter release process was the fusion of the vesicle membrane and the cell membrane, and the transmembrane structural domain of the core protein complex (SNARE) formed a fusion pore in the hydrophobic layer of the fusion membrane, which eventually led to cytosolic spit and transmitter release. Syntaxin3 and synaptosomal neural-associated protein (SNAP-25) were the key proteins of the above membrane fusion process [39,40,41]. The mRNA and protein levels of Syntaxin3 and SNAP-25 in the mouse brain were upregulated to various degrees in the combined fish oil and choline supplementation group. This, in turn, was related to DHA in the brain, and the higher DHA further affected the expression of Syntaxin3 and SNAP-25. Jana et al. found that DHA can increase pairing of Syntaxin3 and SNAP-25 [42]. Syntaxin3 was reported to be essential for neuronal growth, which can be effectively activated by PUFAs crossing the blood–brain barrier. PUFAs were reported to alter the α-helical syntaxin structure to expose the SNARE pattern, allowing SNAP-25 to immediately participate in the process of membrane fusion [43]. Thus, we suspect that the combined intake of fish oil and choline also affects the physiology of cholinergic neurons in the brain and thus affects learning and memory. ACh and choline acetyltransferase (ChAT) is the enzyme responsible for acetylcholine synthesis, and ChAT activity has been shown to be a reliable marker for cholinergic integrity [44]. ACh in the brain altered neuronal excitability, influenced synaptic transmission, and eventually induced synaptic plasticity [45]. Our results showed that both acetylcholine levels and acetylcholine transferase activity were significantly increased in the brains of mice given a combined intake of fish oil and choline, which confirmed our hypothesis.

In conclusion, our study showed that the combined intake of fish oil and choline was, indeed, effective in improving memory in mice (Table 5 and Table 6). Based on our results, we think there are two possible reasons for the effects of fish oil–choline combination supplementation on learning and memory in mice. Firstly, fish oil–choline supplementation promoted brain uptake of DHA in the blood, and the DHA entering the brain affected synaptic plasticity and synaptic function. Secondly, ingested fish oil–choline affected the cholinergic nervous system and accelerated the metabolism of choline itself to synthesize acetylcholine, which participated in the release of neurotransmitters and influenced synaptic transmission and the physiological function of cholinergic neurons. However, which of these is the dominant role is still unknown, and this needs further in-depth study. For our study, we introduced middle-aged mice to our experiments, with a specific focus on studying the effects of intervention on memory in middle-aged individuals. Our findings indicate that this theory is valid in middle-aged C57BL/6 J mice. However, it is important to note that our results only apply to this particular group of mice, so future studies will need to verify whether this conclusion also applies to other mouse strains. Furthermore, it is essential to understand that the effects observed in our experiments may not be applicable for humans. On one hand, humans are more complex than mice; on the other hand, the metabolism of fish oil in humans and mice is also different. Therefore, the improvement of joint intake of fish oil and choline on the content of DHA in brain and the memory still needs to be further confirmed by human clinical trials.

## 5. Conclusions

Our research showed that FC played a significant role in improving the learning and memory ability of C57BL/6 J mice in the water maze experiment and step-through passive avoidance test. After ingesting FC, DHA entered the brain through the BBB in free form or in the form of LPC. The promotion role of FC on the uptake of DHA in the brain was associated with upregulating the expression of fatty acid transporters such as MFSD2A, FATP1, and FABP5 in the brain, as well as the synthesis and release of acetylcholine in the brain. These results suggest that the combined intake of fish oil and choline enhanced the absorption of DHA in the brain and improved the bioavailability of fish oil. This study provides a theoretical basis and technical support for the development of new food nutrition products (such as improving memory products for middle-aged and elderly people).

## Figures and Tables

**Figure 1 foods-12-01799-f001:**
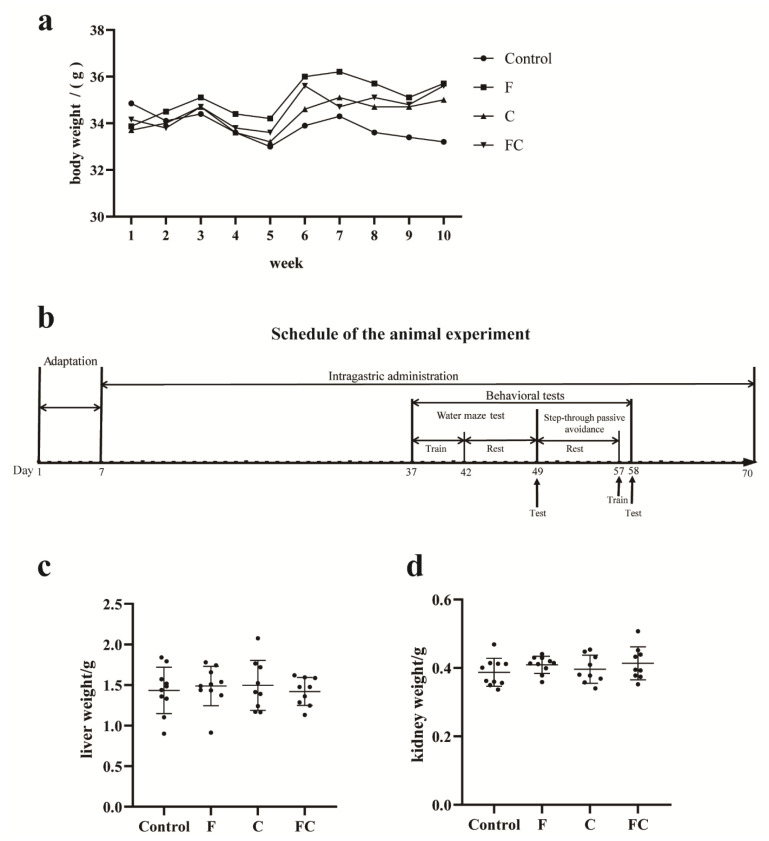
Effects of F, C, and FC on body weight and liver and kidney weight in mice. (**a**) Weekly body weight changes of mice in each group. (**b**) Schedule of the animal experiment. (**c**,**d**) Liver and kidney weights of mice in each group. Values indicate the mean ± SD. Data are presented as mean ± SD values. *n* = 10 for all groups.

**Figure 2 foods-12-01799-f002:**
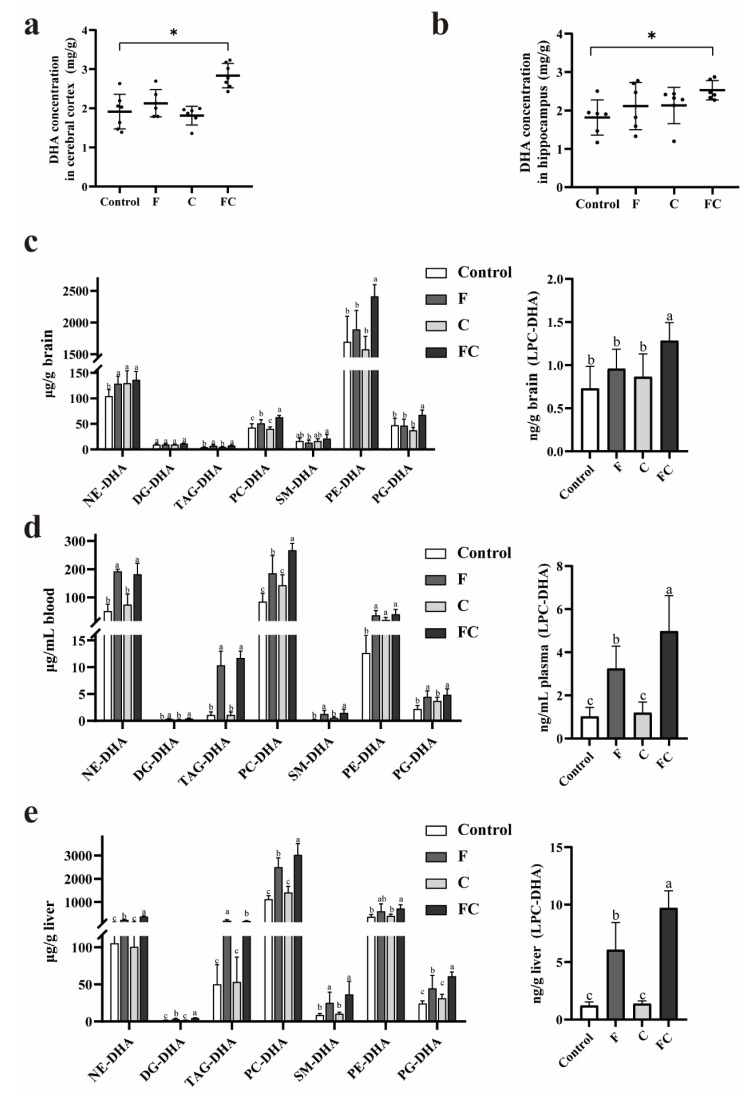
Effects of F, C, and FC on DHA content in brain, plasma, and liver of mice. (**a**) Content of total DHA in the cerebral cortex of each group of mice (mg/g). (**b**) Content of total DHA in the hippocampus of each group of mice (mg/g). Data are presented as mean ± SD values. *n* = 6 for all groups. * Indicates significant difference from the control group (*p* < 0.05). Different lowercase letters indicate significant difference (*p* < 0.05). (**c**) DHA content of different molecules in the brain of mice. (**d**) DHA content of different molecules in the plasma of mice. (**e**) DHA content of different molecules in the liver of mice. Statistical significance between treatments was determined by one-way ANOVA. Bars with common letter superscripts are not significantly different from each other. NE-DHA, nonesterified DHA; DAG-DHA, diacylglycerol-DHA; TAG-DHA, triacylglycerol-DHA; PC-DHA, phosphatidylcholine-DHA; SM-DHA, sphingomyelin-DHA; PE-DHA, phosphatidylethanolamine-DHA; PG-DHA, phosphatidylglycerol-DHA; LPC-DHA, lysophosphatidylcholine-DHA. *n* = 6 for all groups.

**Figure 3 foods-12-01799-f003:**
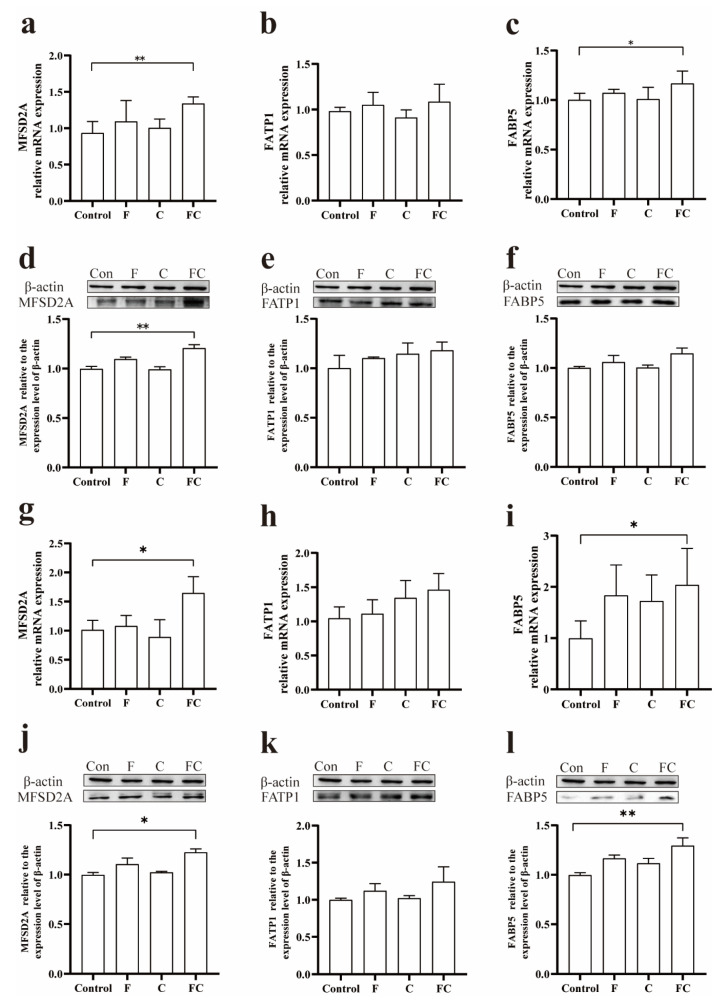
Effects of F, C, and FC on the transport of unsaturated fatty acids. (**a**–**c**), Effects of F, C, and FC on the expression levels of MFSD2A, FATP1, and FABP5 mRNA in mouse cerebral cortex. (**d**–**f**) Effects of F, C, and FC on protein expression levels of MFSD2A, FATP1, and FABP5 in mouse cerebral cortex. (**g**–**i**) Effects of F, C, and FC on the expression levels of MFSD2A, FATP1, and FABP5 mRNA in mouse hippocampus. (**j**–**l**) Effects of F, C, and FC on protein expression levels of MFSD2A, FATP1, and FABP5 in mouse hippocampus. Values indicate the mean ± SD. * Indicates significant difference from the control group (*p* < 0.05). ** Indicates significant difference from the control group (*p* < 0.01). *n* = 3–6 for all groups.

**Figure 4 foods-12-01799-f004:**
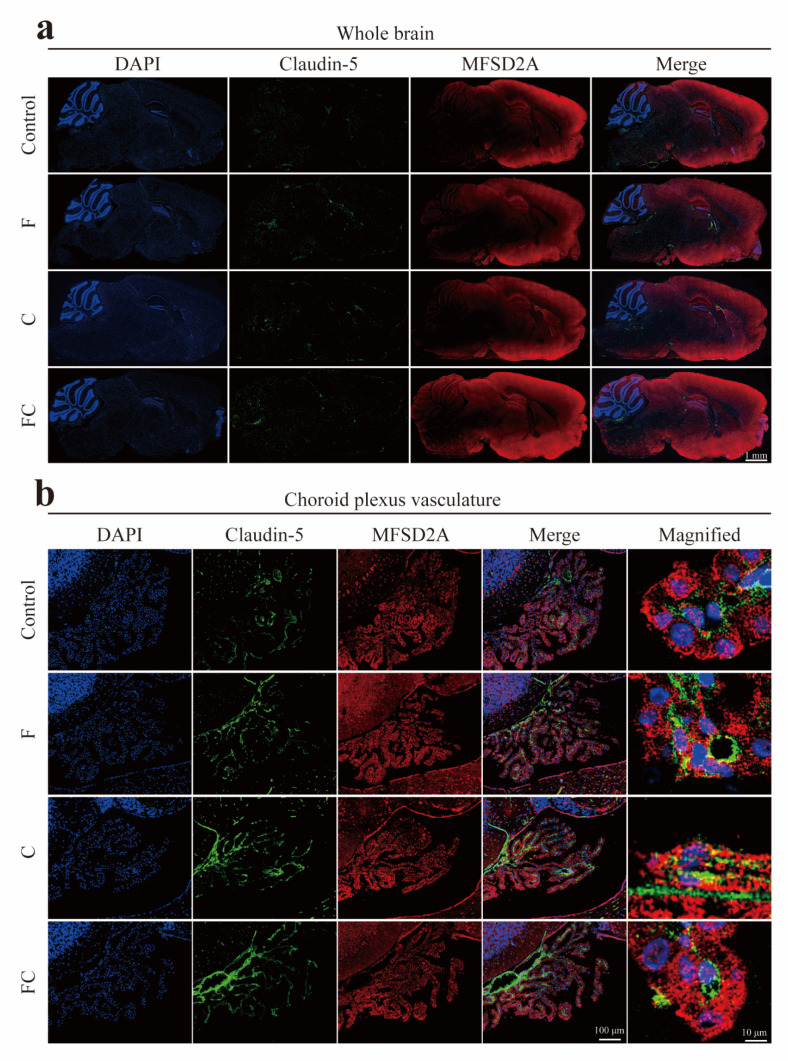
Expression of MFSD2A protein in mouse brain was detected by immunofluorescence double staining. (**a**) Gross morphology and sagittal sections of the brain, immunofluorescence staining of sagittal brain sections from 8-month-old mice of different groups (control, F, C, and FC groups), staining of nuclei using DAPI (shown in blue), localization of vascular endothelial cells using Claudin-5 monoclonal antibody (shown in green), and use of MFSD2A polyclonal antibody to show MFSD2A expression (shown in red). MFSD2A is widely expressed in the brain; scale bar, 1 mm. (**b**) MFSD2A expression in the mouse choroid plexus.; scale bar, 100 μm. Local magnification of the choroid plexus; scale bar, 10 μm. *n* = 3–6 for all groups.

**Figure 5 foods-12-01799-f005:**
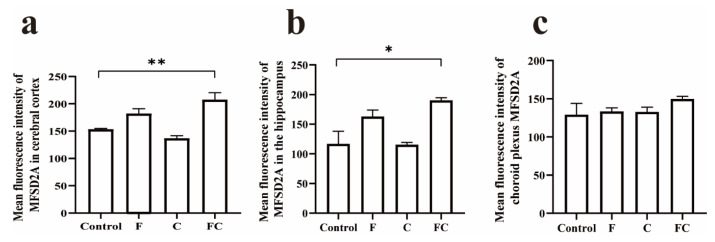
Expression of MFSD2A protein in mouse brain was detected by immunofluorescence double staining. (**a**–**c**) Semiquantitative analysis of immunofluorescence mean fluorescence intensity in the mouse cerebral cortex, hippocampus, and vascular choroid plexus, respectively. Values indicate the mean ± SD. * Indicates significant difference from the control group (*p* < 0.05). ** Indicates significant difference from the control group (*p* < 0.01). *n* = 3–6 for all groups.

**Figure 6 foods-12-01799-f006:**
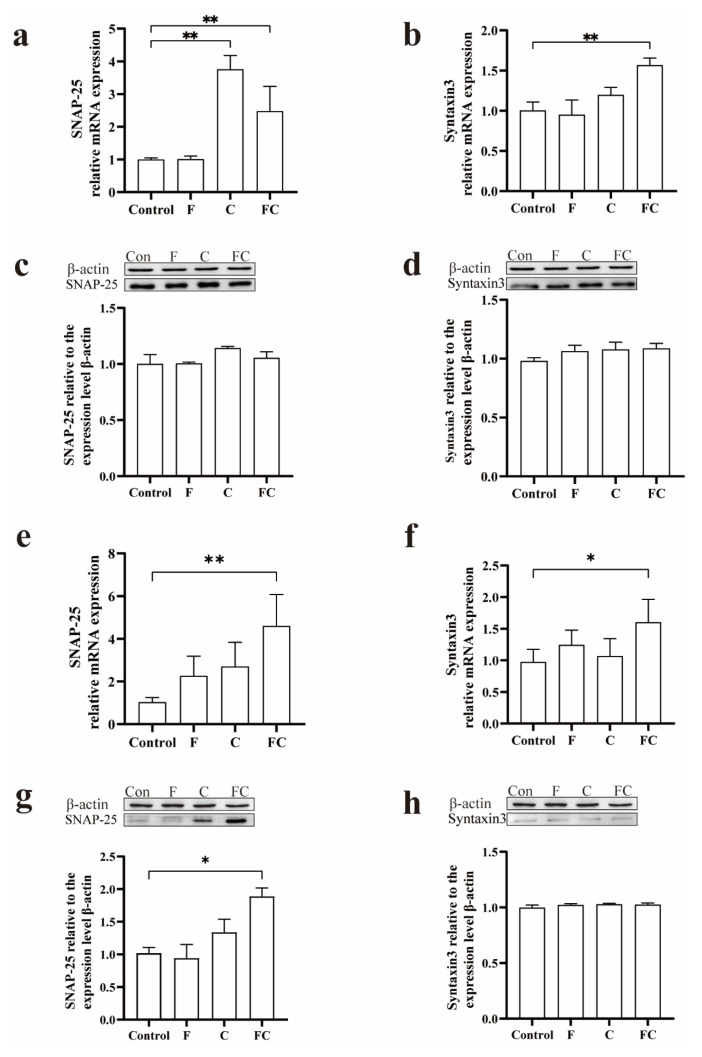
Effects of F, C, and FC on neurotransmitter release. (**a**,**b**) Effects of F, C, and FC on the expression levels of SNAP-25 and Syntaxin3 mRNA in mouse cerebral cortex. (**c**,**d**) Effects of F, C, and FC on protein expression levels of SNAP-25 and Syntaxin3 in mouse cerebral cortex. (**e**,**f**) Effects of F, C, and FC on the expression levels of SNAP-25 and Syntaxin3 mRNA in mouse hippocampus. (**g**,**h**) Effects of F, C, and FC on protein expression levels of SNAP-25 and Syntaxin3 in mouse hippocampus. Values indicate the mean ± SD. * Indicates significant difference from the control group (*p* < 0.05). ** Indicates significant difference from the control group (*p* < 0.01). *n* = 3–6 for all groups.

**Figure 7 foods-12-01799-f007:**
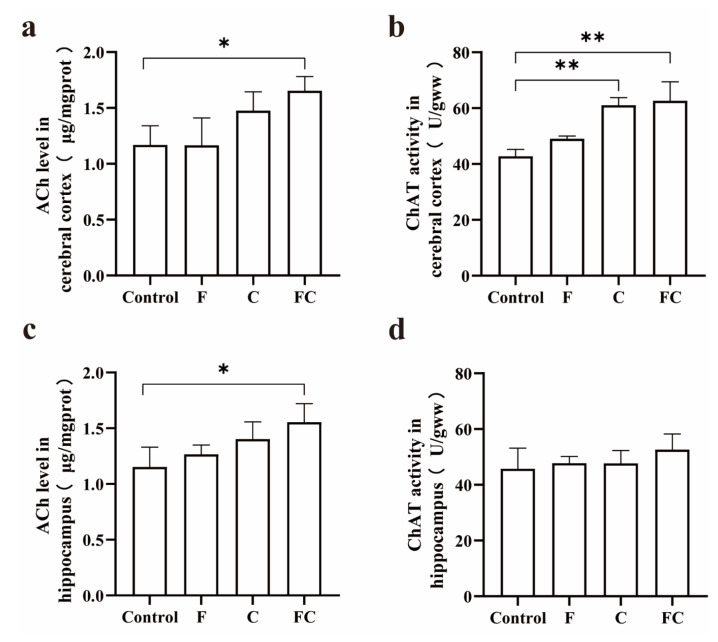
Effects of F, C, and FC on acetylcholine synthesis. (**a**,**b**) Effects of F, C, and FC on ACh content and ChAT activity in mouse cerebral cortex. (**c**,**d**) Effects of F, C, and FC on ACh content and ChAT activity in mouse hippocampus. Values indicate the mean ± SD. * Indicates significant difference from the control group (*p* < 0.05). ** Indicates significant difference from the control group (*p* < 0.01). *n* = 4 for all groups.

**Figure 8 foods-12-01799-f008:**
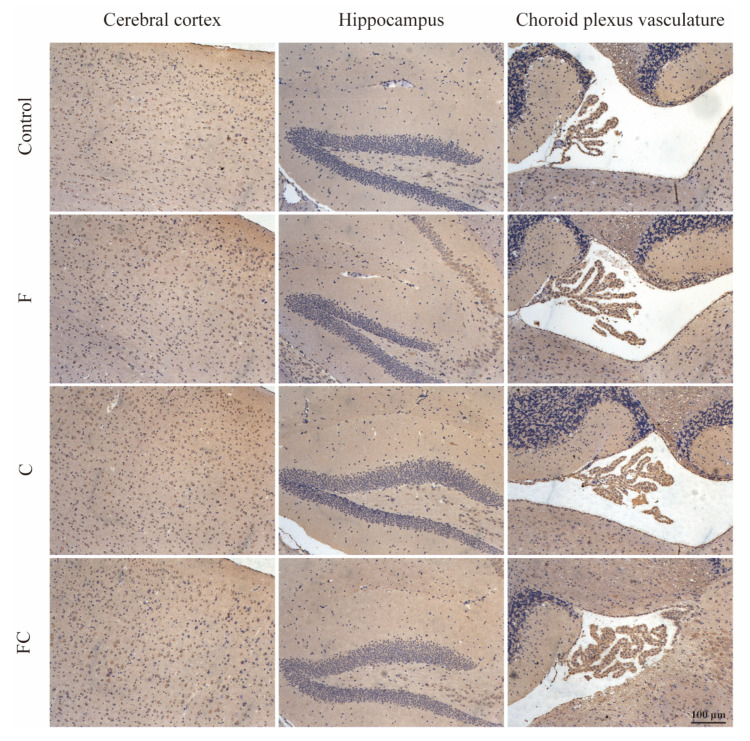
FATP1 expression was analyzed by immunohistochemistry on a panel of paraffin-embedded mouse brain tissue. Sagittal brain sections of 8-month-old mice from different groups (control, F, C, and FC groups): the local field of view of the sections from left to right are the cerebral cortex, hippocampus, and choroid plexus of the fourth ventricle of the mice, respectively. The nucleus of hematoxylin-stained tissue is blue, and the positive expression of DAB is brownish yellow. Scale bar, 100 μm. *n* = 4–5 for all groups.

**Figure 9 foods-12-01799-f009:**
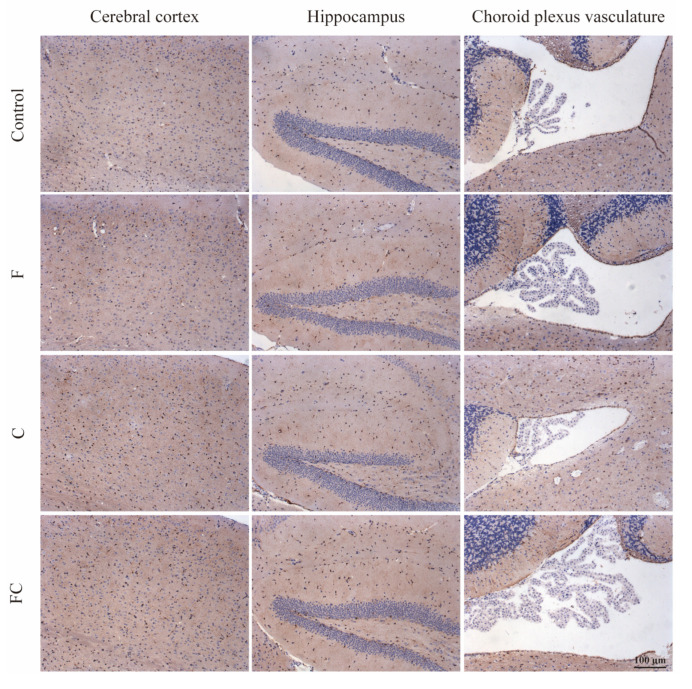
FABP5 expression was analyzed by immunohistochemistry on a panel of paraffin-embedded mouse brain tissue. Sagittal brain sections of 8-month-old mice from different groups (control, F, C, and FC groups): the local field of view of the sections from left to right are the cerebral cortex, hippocampus, and choroid plexus of the fourth ventricle of the mice, respectively. The nucleus of hematoxylin-stained tissue is blue, and the positive expression of DAB is brownish yellow. Scale bar, 100 μm. *n* = 4–5 for all groups.

**Table 1 foods-12-01799-t001:** Rodent diet formulation.

Ingredient	gm%	Kcal%
Casein	200	800
L-Cystine	3	12
Corn starch	425.75	1674
Maltodextrin 10	132	528
Sucrose	100	400
Cellulose, BW200	50	0
Palm stearin	50	450
Mineral Mix S10022G	35	0
Vitamin Mix V10037	3	40
Choline bitartrate	1.25	0
Total	1000	3904
Protein	20	20.8
Carbohydrate	66	67.7
Fat	5	11.5
Total		100

**Table 2 foods-12-01799-t002:** Composition of fish oil and choline.

Raw Material	Functional Ingredients	Content (%)
Fish oil ①	DHA/EPA	52.1%/13.8%
Fish oil ②	DHA/EPA	60.5%/7.56%
Choline bitartrate	Choline	47.40%

A mixture of fish oil ① and fish oil ② was used to obtain fish oil with a 4 to 1 ratio of DHA to EPA.

**Table 3 foods-12-01799-t003:** Feeding dose for each mouse group.

Groups	Feeding Dose ^a^
Control	0.5% carboxymethyl starch solution
Fish oil	0.415 g/kg
Choline	0.14 g/kg
Fish oil and choline	0.555 g/kg

^a^ The volume of intragastric solution fed daily to all mice for 2 months was 9 mL/(kg BW).

**Table 4 foods-12-01799-t004:** Primers used in the measurement of mRNA expression.

Gene	Sequence (5′ to 3′)
MFSD2A	(F) AGAAGCAGCAACTGTCCATTT
	(R) CTCGGCCCACAAAAAGGATAAT
FATP1	(F) CGCTTTCTGCGTATCGTCTG
	(R) GATGCACGGGATCGTGTCT
FABP5	(F) TGAAAGAGCTAGGAGTAGGACTG
	(R) CTCTCGGTTTTGACCGTGATG
Syntaxin3	(F) CTTGATGTACCGGACGCATTC
	(R) ACACTGTCACAATCTGCTCAG
SNAP-25	(F) CAACTGGAACGCATTGAGGAA
	(R) GGCCACTACTCCATCCTGATTAT
β-actin	(F) GGCTGTATTCCCCTCCATCG
	(R) CCAGTTGGTAACAATGCCATGT

**Table 5 foods-12-01799-t005:** Effects among all the groups in enhancing the memory function in the water maze experiment.

Groups	Train (s)		Test (s)	
	Total Time	Total Number of Errors	Time	Number of Errors in 2 Min
Control	335.70 ± 118.03 ^a^	24.20 ± 5.69 ^b^	84.30 ± 38.49 ^a^	6.10 ± 3.22 ^b^
Fish oil	301.46 ± 77.91 ^a^	20.33 ± 4.09 ^b^	77.10 ± 40.15 ^a^	3.70 ± 2.26 ^a^
Choline	300.55 ± 85.95 ^a^	20.78 ± 4.66 ^b^	82.33 ± 41.72 ^a^	3.00 ± 1.22 ^a^
Fish oil and choline	296.45 ± 60.44 ^a^	15.14 ± 3.76 ^a^	70.11 ± 40.56 ^a^	2.63 ± 2.13 ^a^

Data are presented as mean ± SD values. *n* = 10 for all groups. Statistical significance between treatments was determined by one-way ANOVA. Bars with common letter superscripts are not significantly different from each other.

**Table 6 foods-12-01799-t006:** Effects among all the groups in enhancing the memory function in the step-through passive avoidance test.

Groups	Train (s)				Test (s)	
	Latency	Number of Errors	Latency	Number of Errors	Latency	Number of Errors
Control	165.35 ± 74.76 ^a^	1.50 ± 0.71 ^a^	141.57 ± 81.48 ^ab^	1.80 ± 0.92 ^a^	160.26 ± 70.37 ^a^	3.10 ± 0.32 ^b^
Fish oil	75.80 ± 84.31 ^a^	2.00 ± 0.81 ^a^	193.24 ± 101.79 ^ab^	1.60 ± 0.70 ^a^	218.88 ± 95.78 ^a^	1.90 ± 1.67 ^a^
Choline	115.94 ± 83.14 ^a^	1.78 ± 0.67 ^a^	107.92 ± 109.0 ^a^	1.67 ± 0.50 ^a^	174.40 ± 102.44 ^a^	2.44 ± 1.42 ^a^
Fish oil and choline	85.34 ± 62.13 ^a^	1.89 ± 0.60 ^a^	232.34 ± 86.72 ^b^	0.89 ± 0.60 ^b^	208.97 ± 104.96 ^a^	1.67 ± 1.58 ^a^

Data are presented as mean ± SD values. *n* = 10 for all groups. Statistical significance between treatments was determined by one-way ANOVA. Bars with common letter superscripts are not significantly different from each other.

## Data Availability

The data that support the findings of this study are available from the corresponding author upon reasonable request.

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
