# Peer review of "Choline and Fish Oil Can Improve Memory of Mice through Increasing Brain DHA Level"

_foods, 2023, doi:10.3390/foods12091799_

Round 1

Reviewer 1 Report

This manuscript involves studies in mice, The hypothesis was that fish oil + choline in diet (given orally) would improve memory/learning in mice and if so to explore mechanisms.

Several issues need to be addressed by the authors:

1. The authors should acknowledge that fish oil (containing DHA) does increase brain DHA in experimental animals fed on low n-3 PUFA diets (hundreds of studies in the literature).

2. What you are studying is whether it is possible to get a "supra-normal" level of DHA in brain. 

2a. Is it realistic to expect difference in learning/memory between control animals and those with supra-normal DHA levels?

2b. Was the study adequately powered to detect differences in learning/memory? Please provide the power calculation and the biological/behavioural parameter on which you conducted the power calculation.

2c. Please explain the rationale for why 8-month old mice were used.

3. The authors refer to FO1 and FO2, but it is not clear in Table 3 which fish oil was used? Suggest add footnote to table 3 to explain this.

3a. Please indicate where the fish oil was obtained from and whether you conducted analyses to determine if the oil was free from peroxidation. 

4. Lipidomics (sect 2.7): did the authors use standards of different lipids in the analysis?

5. In Figure 2, brain DHA was significantly raised by FO+C treatment. The authors then show brain PC-DHA and brain PE-DHA in Fig 2c. The concentration in PC-DHA was approx 2000ug/g while the concentration of PE-DHA was about 50ug/g (a 40-fold difference). This is contrary to many publications in the literature on DHA in rodent  brain PE and PC where the brain PE-DHA can be as much as 6-times greater than brain PC-DHA (see De Mar JC, J Lipid Res 2006). To explain this further, typically the content of PC and PE in the brain is approx equal but the proportion of DHA in PC is low whereas it is high in PE. Therefore, I query your lipidomic analysis and that is why I asked if you had run standards through your system!

6. Did you measure the choline concentrations in the brain? If not, why not?

7. There should be a paragraph in the Discussion on the strengths and weaknesses of this study.

Author Response

  1. The authors should acknowledge that fish oil (containing DHA) does increase brain DHA in experimental animals fed on low n-3 PUFA diets (hundreds of studies in the literature).

Response 1: Thanks for your comments. We acknowledge that fish oil (containing DHA) did increase brain DHA in experimental animals on a low n-3 PUFA diet, as shown in Line 632. Our experimental results showed that the DHA content in the cerebral cortex and hippocampus of mice after fish oil intake was slightly higher than that of the control group.

  1. What you are studying is whether it is possible to get a "supra-normal" level of DHA in brain.

Response 2: Thanks for your comments. This experiment was conducted on middle-aged and elderly mice, and it appears from the results that it is possible to obtain "supernormal" levels of DHA in the brain, but we must acknowledge that the level of DHA in the brain is regulated by homeostatic mechanisms in the brain and that its level does not increase further.

(References:Lipase treatment of dietary krill oil, but not fish oil, enables enrichment of brain eicosapentaenoic acid (EPA) and docosahexaenoic acid (DHA))

2a. Is it realistic to expect difference in learning/memory between control animals and those with supra-normal DHA levels?

Response 2a: Thanks for your comments.

(References:

Takeyama E,Islam A,Watanabe N, et al. Dietary Intake of Green Nut Oil or DHA Ameliorates DHA Distribution in the Brain of  a Mouse Model of Dementia Accompanied by Memory Recovery.

Green, K.N.; Martinez-Coria, H.; Khashwji, H.; Hall, E.B.; Yurko-Mauro, K.A.; Ellis, L.; LaFerla, F.M. Dietary Docosahexaenoic Acid and Docosapentaenoic Acid Ameliorate Amyloid-β and Tau Pathology via a Mechanism Involving Presenilin 1 Levels.

Yip, P.K.; Bowes, A.L.; Hall, J.C.; Burguillos, M.A.; Ip, T.; Baskerville, T.; Liu, Z.-H.; Mohamed, M.A.; Getachew, F.; Lindsay, A.D. Docosahexaenoic acid reduces microglia phagocytic activity via miR-124 and induces neuroprotection in rodent models of spinal cord contusion injury)

2b. Was the study adequately powered to detect differences in learning/memory? Please provide the on which you conducted the power calculation.

Response 2b: Thanks for your comments. The behavioral experiments we used include Step-through Passive avoidance experiment and water maze experiment, which are currently industry-recognized and commonly used tests for detecting learning-memory abilities, and we refer to the literature for statistical calculations of the experimental results. In behavioral experiments, the longer the latency of the mouse and the less the number of errors of the mouse, the better its learning and memory ability. Compared with the mice in the control group, if there are significant differences in the latency period and the number of mistakes, it means that they are significantly better than the mice in the control group in terms of learning and memory. And our FC group had the best results in behavioral training and testing, p-value less than 0.05.

(References:

AGLPM and QMDDQ peptides exert a synergistic action on memory improvement against scopolamine-induced amnesiac mice.

Ebrahimi-Ghiri M,Rostampour M,Jamshidi-Mehr M, et al. Role of CA1 GABA A  and GABA B  receptors on learning deficit induced by D-AP5 in passive avoidance step-through task.

Micale V,Cristino L,Tamburella A, et al. Enhanced cognitive performance of dopamine D3 receptor "knock-out" mice in the step-through passive-avoidance test: assessing the role of the endocannabinoid/endovanilloid systems.)

2c. Please explain the rationale for why 8-month old mice were used.

Response 2c: Thanks for your comments. The reason why we used 8-month old mice is that we would like to know whether fish oil has effect on healthy middle-aged male mice. Because there is little known about the impact of fish oil in healthy middle-aged male mice. To date, male studies have primarily been completed in disease models and young mice. It is now widely accepted in the industry that aging can lead to memory loss and that early intervention can be made through diet, so more attention is now being paid to the effects of aging on middle-aged people in their 40s due to aging. In the current study, experiments were conducted primarily on middle-aged and older mice to study the effects of DHA intake on memory in this age group. 8-month-old mice eventually reach 10 months of age or older after feeding and fall into the category of middle-aged mice equivalent to the 40-year stage in humans. Therefore, we used 8-month-old mice for the study.

  1. The authors refer to FO1 and FO2, but it is not clear in Table 3 which fish oil was used? Suggest add footnote to table 3 to explain this.

Response 3: Thanks for your comments. A mixture of fish oil â‘  and fish oil â‘¡ is used to obtain fish oil with a 4 to 1 ratio of DHA to EPA, which is more common in the market. Additional explanation has been included in the manuscript, as shown in Line 84-93.

3a. Please indicate where the fish oil was obtained from and whether you conducted analyses to determine if the oil was free from peroxidation.

Response 3a: Thanks for your comments. Our fish oil is from Novosana and we performed a peroxide test prior to the experiment and the result was 8.3 meq/kg. which met the experimental criteria.

  1. Lipidomics (sect 2.7): did the authors use standards of different lipids in the analysis?

Response 4: Thanks for your comments. Yes, different standards were used for each lipid, and a mixed-label product SPLASH Lipidomix (330707) from Avanti Polar Lipids was used

  1. In Figure 2, brain DHA was significantly raised by FO+C treatment. The authors then show brain PC-DHA and brain PE-DHA in Fig 2c. The concentration in PC-DHA was approx 2000ug/g while the concentration of PE-DHA was about 50ug/g (a 40-fold difference). This is contrary to many publications in the literature on DHA in rodent brain PE and PC where the brain PE-DHA can be as much as 6-times greater than brain PC-DHA (see De Mar JC, J Lipid Res 2006). To explain this further, typically the content of PC and PE in the brain is approx equal but the proportion of DHA in PC is low whereas it is high in PE. Therefore, I query your lipidomic analysis and that is why I asked if you had run standards through your system!

Response 5: Thank you for pointing out the problem, later we re-examined and recalculated and found that the grouping of PE-DHA and PC-DHA was wrong, which has been changed now, but our data showed that PE-DHA was much higher than PC-DHA, which may be due to the difference of mouse species and age, experimental environment and assay method. Finally, thank you again for your feedback, and we will continue to focus on this issue in future experiments.

  1. Did you measure the choline concentrations in the brain? If not, why not?

Response 6: Thanks for your comments. We did not test the concentration of choline in the brain because after choline intake, choline is metabolized in vivo to methylated metabolites or acetylcholine, and the level of acetylcholine synthesis is an important indicator of the cholinergic system, which plays an important role in the learning and memory processes. In numerous related literatures, after choline ingestion in mice, the level of acetylcholine is mainly tested, as well as some indicators affecting acetylcholine synthesis, such as choline acetyl transferase or acetylcholinesterase, and the choline content in the brain is not retested.

(References:

AGLPM and QMDDQ peptides exert a synergistic action on memory improvement against scopolamine-induced amnesiac mice)

  1. There should be a paragraph in the Discussion on the strengths and weaknesses of this study.

Response 7: Thanks for your suggestion. We have added the strengths and weaknesses of this study to the discussion, as shown in Line355-364.

Reviewer 2 Report

Docosahexaenoic acid(DHA) is an important component of brain lipids and that of central nervous system. Deficiency in the accumulation of DHA in the brain lipids was perceived to affect memory and learning abilities of human subjects ,particularly in growing children.. Circumstantial evidences indicate that less than adequate amounts of DHA in brain lipids may lead to neurological disorders. The uptake of circulating DHA by brain cells has to overcome Blood Brain Barrier. Hence several attempts are made to boost the levels of DHA by using nano-technology, co delivering fish oil with phenolic like curcumin from the spice turmeric. In the present manuscript , the authors have used fish oil as source of DHA and supplementing it with choline (a component of phosphatidylcholine).These formulations were fed to mice for two months by gavage. Mice were monitored for DHA accumulation, memory, learning abilities and changes at molecular levels of relevant genes. The results indicated that intake of fish oil with choline improved DHA status and higher scores for learning ability and in memory tests The fish oil-choline combinations influenced cholinergic nervous system and acetylcholine levels. The fish oil -choline combinations upregulated the expression of fatty acid transporters. Based on these results, the authors concluded that combined intake of fish oil with choline improved the bioavailability of DHA and positively influenced cognitive functions in mice.

Clarifications required:

1. What was the source of fish oil which had 52-61% DHA and 7.56 to 13.8% EPA (Table 2)

2.The over all fatty acid composition of the dietary fats may be given. Did the diet contain any alpha-linolenic acid?

Is there any in situ synthesis of DHA containing lipids in brain tissue, which may be stimulated by choline? Please comment

Author Response

  1. What was the source of fish oil which had 52-61% DHA and 7.56 to 13.8% EPA (Table 2)

Response 1: Thanks for your comments. Our fish oil is from Novosana. A mixture of fish oil â‘  and fish oil â‘¡ is used to obtain fish oil with a 4 to 1 ratio of DHA to EPA, which is more common in the market. Additional explanation has been included in the manuscript, as shown in Line 84-93.

2.The overall fatty acid composition of the dietary fats may be given. Did the diet contain any alpha-linolenic acid?

Response 2: Thanks for your comments. The 4% fat source in the feed is palm stearin. The range of alpha-linolenic acid detected in the diet is 0-0.5%. Almost no alpha-linolenic acid was present. The following are the levels of other fatty acids in the feed.

C22:6: ND~0.5%

C20: 5: ND~0.5%

C18:3: ND~0.5%

C18:1: 15~36%

C18:2: 3~10%

C12: 0: 0.1~0.5%

C14: 0: 1~2%

C16:0: 48~74%

C18:0: 3.9~6%

3.Is there any in situ synthesis of DHA containing lipids in brain tissue, which may be stimulated by choline? Please comment

Response 3: Thanks for your comments. Some studies suggest that DHA can be synthesized in the brain from its precursor, alpha-linolenic acid (ALA), which is an essential fatty acid that must be obtained through the diet. Other studies suggest that DHA can also be obtained from the diet and transported into the brain. Our findings suggest that choline supplementation may increase the brain's intake of DHA and does not stimulate the in situ synthesis of DHA containing lipids in the brain.

Round 2

Reviewer 1 Report

The authors have responded to my previous comments, but hardly changed the manuscript at all, so my suggestions have not been incorporated into the ms.

They must incorporate these into the revised version, or argue why they do NOT need to change the ms.

Issues to be addressed:

Power calculation,

Rationale for using 8-month old mice (are these elderly mice?)

Oxidation status of PUFA used,

Standards used in the analysis of the lipids

Author Response

Thanks for your comments.We have studied your comments carefully and have made revision which marked up using the “Track Changes” in the paper. We have tried our best to revise our manuscript according to the comments. Attached please find the revised version, which we would like to submit for your kind consideration.We would like to express our great appreciation to you for comments on our paper.

1.Power calculation

Response : We would like to thank the reviewer for the suggestions on our revised manuscript.

As suggested by the reviewer, we modified the description , which is in line113~114, 195~162, 205~213, 216~219, and 225~226:

“Sample size for experiments was decided based on power calculation from previous experiments and experiences.”

“The statistical power (power) of the behavioral experiments was calculated by G∗Power 3.1.9.2 software with an α-error probability value of 0.05.”

“Behavioral tests are commonly used in nonclinical effectiveness studies to evaluate the impact of drugs or other interventions on learning and memory function in animals. The behavioral experiments we used include Step-through Passive avoidance experiment and water maze experiment, which are currently industry-recognized and commonly used tests for detecting learning-memory abilities. Sample size for experiments was decided based on power calculation from previous experiments and experiences. Statistics and calculations of behavioral experimental results were based on previous studies with modifications. In behavioral experiments, the longer the latency of the mouse and the less the number of errors of the mouse, the better its learning and memory ability.”

“Mice in the FC group made significantly fewer errors in the water maze training than the control group (P<0.05, power=0.976). Similarly, in the water maze test experiment, the number of errors in the FC group of mice was significantly less than the control group (P<0.05, power=0.8336).”

“In the step-through passive avoidance test experiment, the number of errors in FC mice was significantly lower than in the control group (P<0.05, power=0.951).”

  1. Rationale for using 8-month old mice (are these elderly mice?)

Response: We would like to thank the reviewer for the suggestions on our revised manuscript.

They are middle-aged mice.

As suggested by the reviewer, we modified the description , which is in line 73~83:

“There is little known about the impact of fish oil in healthy middle-aged male mice. To date, male studies have primarily been completed in disease models and young mice. It is now widely accepted in the industry that aging can lead to memory loss and that early intervention can be made through diet, so more attention is now being paid to the effects of aging on middle-aged people in their 40s due to aging. In the current study, experiments were conducted primarily on middle-aged and older mice to study the effects of DHA intake on memory in this age group. 8-month-old mice eventually reach 10 months of age or older after feeding and fall into the category of middle-aged mice equivalent to the 40-year stage in humans. Therefore, this study used 8-month-old mice to investigate whether the combined intake of fish oil and choline significantly improved memory in mice and to investigate the mechanism of why the combined intake of fish oil and choline affected memory in mice.”

  1. Oxidation status of PUFA used

Response: We would like to thank the reviewer for the suggestions on our revised manuscript.

As suggested by the reviewer, we modified the description , which is in line 97~98:

“Our fish oil is from Novosana and we performed a peroxide test prior to the experiment and the result was 8.3 meq/kg. which met the experimental criteria.”

4.Standards used in the analysis of the lipids

Response: We would like to thank the reviewer for the suggestions on our revised manuscript.

As suggested by the reviewer, we modified the description , which is in line 154~155:

“We used different standards for each lipid and used the mixed-label product SPLASH Lipidomix (330707) from Avanti Polar Lipids.”
